# Representing Agentic Tools in Knowledge Graphs for Structure-Aware Tool Discovery Under Tool Overload

Isaiah Onando Mulang'[1], Johannes Thaller[1], Tushar Trivedi[1], Lars Heling[1] and Felix Sasaki[1]

[1]*SAP SE, Dietmar-Hopp-Allee 16, 69190 Walldorf, Germany*

## Abstract

Large language model (LLM) agents increasingly rely on external tools, yet most tool ecosystems still expose those tools as unstructured textual descriptions or JSON schemas. As tool inventories grow, this becomes a retrieval problem where the agent must surface a small relevant set under context and tool-budget constraints. We study knowledge-graph-based tool representation for agentic systems through a lightweight ontology for Model Context Protocol (MCP) tools. The ontology models tools, servers, capabilities, and parameters, and treats required versus optional inputs as first-class relations. Using real MCP tool schemas extracted from publicly available servers, we build an RDF knowledge graph. We instantiate this knowledge graph on MCP-Atlas, a benchmark for tool-use competency built around real MCP servers, and compare a KG-augmented discovery workflow against a text-only baseline across multiple frontier models and two exposure regimes: the benchmark's smaller task-level tool menus and an overload setting with an all-tools registry of approximately 269 tools over 258 executed tasks. The early empirical results show specific and actionable insights. In smaller curated tool settings, direct text-only exposure of names, descriptions, and JSON schemas remains stronger for all tested models. However, under overload where the unstructured baseline is constrained by a maximum tool budget, KG-based filtering improves GPT-5 from 0.478 to 0.542 mean coverage. For Claude 4.6 Sonnet in the all-tools condition, the KG retains roughly 89% of the text baseline's coverage while reducing the candidate set from about 270 tools to 4.6 tools on average. Qualitative error analysis indicates that the KG helps primarily by reducing tool overload, name ambiguity, and backend confusion, while its main weakness is incomplete recall caused by missing or imperfect capability assignments. The central conclusion validates the value of knowledge graphs as a structure-aware compression layer for large, noisy tool registries, and opens a larger research question on best approaches to represent tools and knowledge graphs together with strong textual and JSON-tool descriptions.

## Keywords

knowledge graphs, tool discovery, MCP, agentic AI, ontology engineering, tool use, LLM agents

## 1. Introduction

Tool use has become a core capability of modern LLM agents [1, 2, 3]. Instead of answering from parametric memory alone, agents are designed to search documents, query databases, call APIs, read files, or compose multiple tools into a task-specific workflow that meets a target goal. An incipient research direction for tool use in agentic settings follows multi-tool orchestration over long trajectories [3], which in turn demands efficient discovery of the tools in use. In practice, however, tool interfaces are still mostly exposed as flat lists of names, descriptions, and JSON schemas. This is workable at small scale, but it becomes brittle once agents face tens or hundreds of overlapping, potentially ambiguous tools across multiple servers and domains. Concomitantly, Knowledge Graphs [4, 5] have been established as an authoritative way to structure information in the enterprise and for consumption by generative models through ideas such as graph-retrieval-augmented generation (Graph-RAG) [6, 7] or as encoded graph tokens [8, 9, 10] imbibed into large language models, and there exists a substantial body of work on efficient graph representation, querying, and reasoning.

The overarching research question in this work is to investigate whether a Knowledge Graph (KG) can serve as a better representation layer for tool discovery, anchored on the central idea that tool selection is not only a lexical matching problem. It is also a structural reasoning problem involving

*GENAIK-NORA 2026: Joint Workshop on Generative AI and Knowledge Graphs and Knowledge Graphs & Agentic Systems Interplay, IJCAI-ECAI 2026 Workshops, August 2026, Bremen, Germany*

✉ mualang.onando@sap.com (I. O. Mulang'); johannes.thaller@sap.com (J. Thaller); tushar.trivedi@sap.com (T. Trivedi); lars.heling@sap.com (L. Heling); felix.sasaki@sap.com (F. Sasaki)

capability hierarchies, server provenance, parameter semantics, and constraints on how tools can be used. A Knowledge Graph makes these structures explicit, supports transparent traversal and validation, and reduces the number of tools that must be surfaced to the downstream agent.

Although there are numerous ways to represent and serve tools for agentic use, we focus on the Model Context Protocol (MCP) [11, 12] because it has emerged as a practical interoperability layer for agentic tools and because MCP-Atlas provides a community-relevant benchmark built on real servers, controlled distractor tools, multi-step workflows, and claims-based evaluation [13]. Our ontology centers on four core classes: *Server*, *Tool*, *Capability*, and *Parameter*. We map MCP schemas into RDF triples, validate the graph with SPARQL, and use it as a discovery layer before execution. Our evaluation supports a more precise claim than "KGs outperform text." The current KG helps mainly in overload regimes, where the model must choose under too many tools, semantically overlapping descriptions, or provider-specific tool limits. In smaller curated settings, unstructured text and JSON schemas remain better because they preserve recall and avoid an additional routing stage. This distinction matters for the GenAIK-NORA audience: the contribution is not only an ontology, but also an empirical characterization of when symbolic structure helps agentic tool use and when it does not.

This paper makes four contributions. First, it introduces a lightweight ontology for MCP-style tools that models servers, tools, capabilities, and parameters, while explicitly distinguishing required from optional inputs. Second, it describes a KG construction workflow from real MCP tool schemas to RDF, with validation queries for graph integrity and relation traversal. Third, it presents a comparative evaluation of KG-augmented tool discovery versus text-only exposure of tool descriptions and JSON schemas across multiple models and tool-budget regimes. Fourth, it offers a fine-grained analysis showing that the KG's main benefit is structure-aware filtering under tool overload, whereas its main failure mode is recall loss from incomplete capability coverage, and it outlines concrete directions to mitigate this recall limitation.

## 2. Background and Related Work

### 2.1. Tool Schemas and Tool-Use Evaluation

Recent agent ecosystems have converged on schema-based tool descriptions, typically centered on JSON-style parameter specifications. Practitioner guidance from Anthropic emphasizes that effective tools need clear names, high-signal descriptions, token-efficient responses, and explicit evaluation, because agents are easily confused by overlapping or overly generic tool interfaces [14]. Likewise, tool-schema engineering guides describe JSON Schema as the de facto foundation for specifying name, description, parameter types, required fields, defaults, and constraints [15]. These sources are vital since our ontology is intentionally grounded in the fields that appear consistently across real tool schemas. In our text baseline, these JSON schemas and descriptions are flattened into natural-language context for the model, while the KG-based approach reuses the same underlying schemas structures as a graph.

Benchmarking work has moved from isolated function-calling tasks toward more realistic agentic evaluation. Berkeley Function Calling Leaderboard (BFCL) evaluates tool and function calling performance across diverse scenarios and has evolved from function-call accuracy toward broader agentic evaluation [16]. ToolSandbox argues that realistic benchmarking requires stateful execution, implicit dependencies between tools, and conversational interaction [17]. Tool Playgrounds and StableToolBench likewise highlight the need for large-scale, analyzable, and stable evaluation environments for tool-using agents [18, 19]. Most directly relevant to this paper, MCP-Atlas evaluates tool-use competency with real MCP servers, multi-step workflows, controlled tool exposure, distractors, and claims-based scoring [13]. The public release contains 500 tasks, while the public leaderboard evaluates 1,000 tasks across 36 servers. Our evaluation sits within this broader trend, but asks a different question: whether structured tool representation changes which tools are discovered and therefore which tasks are solvable.

## 2.2. Semantic Service Discovery and Composition

The semantic-web community studied automated service discovery long before LLM agents. OWL-S and related work argued that machine-interpretable service descriptions are required for automatic matching, composition, and invocation of services [20, 21]. Other work extended semantic matching with non-functional criteria such as quality-of-service and later adapted ontology-driven discovery to cloud services [22, 23]. These works are clear predecessors to tool discovery for LLM agents. However, classical semantic service discovery targeted web services and enterprise integration, not the interactive, prompt-driven, context-limited behavior of modern LLM agents. Our work revisits the same semantic discovery problem, but under a new operating constraint: agents must reason over tool interfaces using limited context windows and imperfect natural-language plans.

## 2.3. Knowledge Graphs for Agent and Tool Retrieval

Recent work has begun to combine graph-based retrieval with MCP-style agent ecosystems. Agent-as-a-Graph represents tools and parent agents as nodes in a knowledge graph and reports improvements in Recall@5 and nDCG@5 on a live MCP benchmark [24]. That work is closely aligned with the present paper, but focuses on graph-based retrieval for multi-agent systems rather than ontology design for tool schema semantics or a controlled comparison against direct text and JSON exposure. More broadly, graph-based reasoning has been used in domain-specific agent systems such as SciAgents, where ontological graphs help organize concepts and support multi-agent reasoning [25]. Our work differs in scope: it targets the representation and retrieval of executable tools themselves, rather than agents, and emphasizes the trade-off between precision and recall in discovery.

# 3. Problem Statement and Formalization

Let $\mathcal{S}$ be a set of servers, $\mathcal{T}$ a set of tools, $\mathcal{C}$ a set of capabilities, and $\mathcal{P}$ a set of parameters. We model the tool ecosystem as a typed directed graph:

$$\mathcal{G} = (\mathcal{V}, \mathcal{E}, \tau_V, \tau_E),$$

where

$$\mathcal{V} = \mathcal{S} \cup \mathcal{T} \cup \mathcal{C} \cup \mathcal{P}$$

and $\tau_V$ and $\tau_E$ assign node and edge types. The graph contains at least the following edge families:

$$E_{\text{host}} \subseteq \mathcal{T} \times \mathcal{S} \to hostedOn,$$
$$E_{\text{cap}} \subseteq \mathcal{T} \times \mathcal{C} \to hasCapability,$$
$$E_{\text{req}} \subseteq \mathcal{T} \times \mathcal{P} \to hasRequiredInput,$$
$$E_{\text{opt}} \subseteq \mathcal{T} \times \mathcal{P} \to hasInput,$$
$$E_{\text{sub}} \subseteq \mathcal{C} \times \mathcal{C} \to isSubCapabilityOf.$$

Given a task instance $x \in \mathcal{X}$ with natural-language request $q(x)$, a discovery policy $D$ returns a candidate set of tools

$$R_D(x) \subseteq \mathcal{T}, \quad |R_D(x)| \leq B,$$

where $B$ is the tool budget that can be passed to the execution agent. An execution agent $A$ then uses only the candidate set $R_D(x)$ to produce an answer $\hat{y}(x)$ and possibly a tool-use trace $\pi(x)$. Let $y^*(x)$ denote the reference answer and let

$$\text{Cov}(\hat{y}(x), y^*(x)) \in [0, 1]$$

be the benchmark coverage score used by the evaluation harness. The system objective is to maximize expected task coverage:

$$J(D, A) = \mathbb{E}_{x \sim \mathcal{X}} \left[ \text{Cov}(\hat{y}(x), y^*(x)) \right].$$

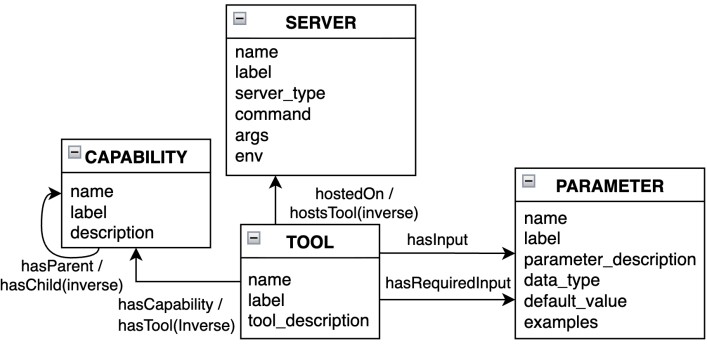

**Figure 1:** The data model: core ontology used in the KG prototype.

This setup makes the core trade-off explicit. A text-only strategy $D_{\text{text}}$ can have high recall because it exposes many or all tools directly as flattened descriptions and JSON schemas, but it risks overload when $B$ is large or when the model must reason over many overlapping descriptions. A KG-based strategy $D_{\text{kg}}$ can reduce the candidate set by exploiting structure in $\mathcal{G}$, but it may fail if the graph omits relevant capabilities or routes the query to the wrong subgraph. To analyze discovery quality more directly, let $T^*(x) \subseteq \mathcal{T}$ denote the oracle set of tools sufficient for solving task $x$. Then discovery precision and recall are

$$\text{Prec}_D(x) = \frac{|R_D(x) \cap T^*(x)|}{|R_D(x)|}, \qquad \text{Rec}_D(x) = \frac{|R_D(x) \cap T^*(x)|}{|T^*(x)|}.$$

The evidence in our experiments indicates that the current KG achieves very high precision but insufficient recall. This position indicates that the KG is good at filtering out irrelevant tools, but not yet good enough at recovering all relevant tools for hard multi-step tasks. In Sections 7 and 8, we return to these precision and recall definitions when interpreting the empirical findings.

## 4. Ontology and Knowledge Graph Construction

### 4.1. Ontology Scope

We intentionally scope the ontology to the entities that matter most for discovery on MCP-style benchmarks. The current graph models four entity types:

- **Server**: the MCP server that hosts one or more tools.
- **Tool**: the main executable entity, described by a name, label, and natural-language description, together with a JSON-schema parameter definition.
- **Capability**: an abstract functional class used for semantic grouping and retrieval.
- **Parameter**: reusable node describing an input argument with type and default value when available.

Figure 1 shows the implemented core class diagram. One noteworthy design choice is that requirement status is modeled relationally rather than as a Boolean attribute on the parameter alone. A parameter may be required for one tool and optional for another, so the distinction belongs naturally to the edge type: *hasRequiredInput* versus *hasInput*. This keeps the ontology closer to the semantics of invocation. The graph is tool-centric, where the relation is modeled as *Tool → hostedOn → Server*, because the tool is the primary retrieval object and this direction eases extension to future non-MCP tool types such as REST APIs or local tools, but we keep a reverse relation in the implementation for convenience. Scope-wise, tool outputs are excluded from the current core ontology. Whereas MCP input schemas are standardized, output schemas are often runtime-dependent or absent. Output modeling, execution constraints, dependencies, and side effects are therefore deferred to future work, as they require richer semantics than what is currently standardized for MCP tools.

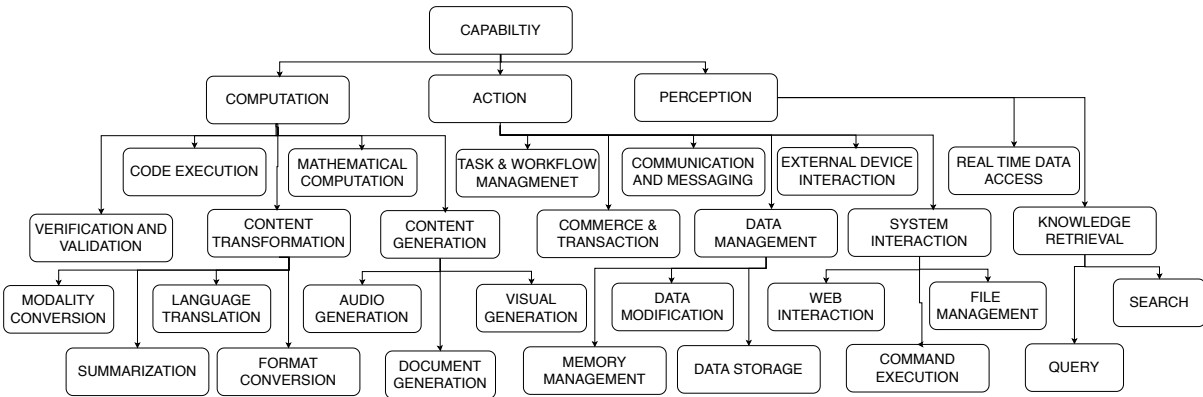

**Figure 2:** Capability taxonomy used to organize tool functionality into retrieval-relevant semantic classes.

## 4.2. Capability Taxonomy

Capabilities are the semantic bridge between raw tool schemas and higher-level tool discovery. Tool selection is driven primarily by what a tool can accomplish, not by the surface form of its name. We therefore organize capabilities into a lightweight hierarchy that reflects recurring functional categories across contemporary tool ecosystems and benchmarks. The current taxonomy includes top-level clusters such as information access, content generation, data processing, and system interaction, with leaf capabilities including web search, database querying, external API access; text, code & image generation, information transformation, computation, file management, and shell execution.

Figure 2 shows the capability taxonomy used for this clustering. This taxonomy serves two roles. First, it supports retrieval by grouping tools that are semantically related even when their names differ. Second, it provides an abstraction layer that can eventually align MCP-native tools with alternative representations such as OpenAPI or other agent-tool registries. While the taxonomy is intentionally lightweight to keep annotation costs manageable, its incompleteness is one of the main contributors to recall loss, because tools whose functionality is not yet captured by a capability node are excluded from graph-based discovery.

## 4.3. Capability Assignment Procedure

Capability assignments connect individual tools to the capability taxonomy. In the current prototype, these assignments are obtained through a weakly supervised, hybrid process. For each MCP tool, we consider its name, description, and JSON-schema parameter metadata as a joint description. A small internal tagging pipeline first proposes one or more candidate capabilities for that tool based on heuristic keyword patterns and semantic similarity to capability labels. In a second step, developers review and correct assignments for tools on selected servers, focusing especially on high-frequency or high-impact capabilities. This hybrid procedure reflects a typical practical trade-off: fully manual annotation would be more accurate but too costly at scale, while fully automated assignment would amplify classification errors and taxonomy gaps.

We did not perform a full quantitative study of capability-assignment quality, for example through inter-annotator agreement or systematic error-rate estimation. However, the error analysis in Section 7 provides indirect evidence about the impact of assignment quality. Tasks that fail in the KG-based setting often involve missing or incorrect capability links for tools that the text baseline can still discover through their flattened descriptions and schemas. In future work, we plan to replace the current heuristic-plus-review pipeline with a more systematic approach based on multi-label classifiers or LLM-assisted tagging with explicit quality control, and to report assignment quality metrics explicitly.

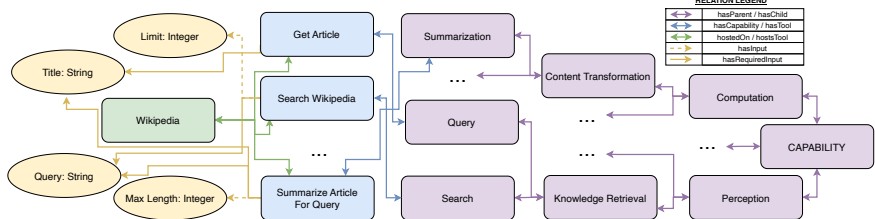

**Figure 3:** Instance-level validation of the ontology using Wikipedia MCP tools. The example highlights server, tool, capability, and parameter mappings.

## 4.4. ETL Pipeline and Validation

The KG is built from real MCP tool schemas in the MCP-Atlas environment [13]. We initially validated the mapping on representative servers such as Wikipedia, whose tools include search, retrieval, summarization, and section-level operations. The same mapping procedure is then applied across benchmark servers and checked with SPARQL queries for relation traversal, inverse consistency, hierarchy traversal, and basic data integrity. The implemented mapping strategy is intentionally lightweight:

1. Extract raw MCP tool schemas from server manifests or JSON outputs.

2. Normalize tool names, descriptions, server metadata, and JSON-schema parameter fields.

3. Create parameter nodes with type and default metadata.

4. Attach parameters using either *hasRequiredInput* or *hasInput*.

5. Map each tool to one or more capabilities from the taxonomy.

6. Serialize the resulting graph to RDF and validate it using SPARQL.

In the current system, only tools with assigned capabilities are included in the KG. This choice improves precision, because unsupported tools are excluded, but it also creates a recall bottleneck whenever relevant tools have not yet been assigned a capability. Figure 3 shows an instance-level Mermaid rendering used to validate mappings for representative tools from the Wikipedia server.

## 5. KG-Augmented Tool Discovery Workflow

The KG-augmented workflow separates *discovery* from *execution*. Instead of presenting the downstream agent with all tool descriptions and schemas up front, the system first queries the graph to retrieve a small candidate subset based on task semantics, capability associations, and structural metadata. The execution agent then operates over that reduced tool set. At a high level, the workflow is as follows. The system parses the task request and identifies likely capabilities, data sources, or action types. It then traverses the graph to retrieve tools linked to those capabilities and their hosting servers. For the tools retrieved in this way, the system inspects the tool descriptions and parameter schemas to select a small number of candidates. The final candidate set is returned to the execution agent, which can then execute one or more MCP tools and produce the final answer.

This is qualitatively different from a text-only baseline that loads all available tools or a selected subset of them directly into the agent context as a flat list of natural-language descriptions and JSON schemas. The KG workflow performs explicit candidate compression before the model commits to execution. In the all-tools regime, this reduces the average candidate set from roughly 270 tools to approximately 4.6 tools in the Claude 4.6 Sonnet setting. Such compression is valuable because practical tool use is constrained not only by retrieval quality but also by context limits, token budgets, and provider-specific caps on the number of tools that can be supplied at once. While we did not log per-request token consumption or latency in our experiments, this reduction in candidate tools suggests that KG-based filtering should translate into shorter tool menus, lower token usage for tool metadata, and potentially better latency and time-to-first-token. Measuring these efficiency metrics explicitly is an important

**Table 1**

Mean coverage scores for KG-based discovery versus text-only tool exposure. Positive Δ means the KG is better.

| Model | Setting | Text | KG | Δ |
|---|---|---|---|---|
| Claude 4.6 Sonnet | Selected tools | 0.778 | 0.713 | -0.065 |
| GPT-5 | Selected tools | 0.658 | 0.584 | -0.074 |
| Gemini 2.5 Pro | Selected tools | 0.201 | 0.088 | -0.113 |
| Claude 4.6 Opus | Selected tools | 0.871 | 0.814 | -0.057 |
| Claude 4.6 Sonnet | All tools | 0.645 | 0.575 | -0.070 |
| GPT-5 | All tools | 0.478 | 0.542 | +0.064 |
| Gemini 2.5 Pro | All tools | 0.152 | 0.102 | -0.050 |
| Claude 4.6 Opus | All tools | 0.764 | 0.653 | -0.111 |

direction for future work.

# 6. Experimental Setup

## 6.1. Benchmark and Protocol

We evaluate on MCP-Atlas, a large-scale benchmark for tool-use competency with real MCP servers [13]. MCP-Atlas is designed for realistic agentic workflows rather than isolated function calls: tasks are written in natural language, typically require three to six tool calls, avoid naming the target tool directly, and are scored with a claims-based coverage metric. Each benchmark instance provides a prompt, a task-specific enabled tool menu, ground-truth claims, and a tool-call trajectory for diagnosis. The public release contains 500 tasks, while the leaderboard evaluates 1,000 tasks across 36 servers spanning search, analytics, productivity, finance, and coding. The reported runs use the MCP-Atlas environment and scoring methodology. Under the active server configuration for our experiments, the executable benchmark slice covers 258 tasks. In the overload regime, the accessible registry expands to approximately 269 tools, which is large enough to expose the retrieval bottleneck that this paper targets.

## 6.2. Compared Conditions

We compare two retrieval settings. In the *selected-tools setting*, which is closest to the standard MCP-Atlas protocol, the execution agent receives a smaller task-level tool menu rather than the full registry. This setting tests whether KG mediation still helps when tool overload is already controlled by curated menus. In the *all-tools setting*, we construct an additional stress test beyond the default benchmark configuration. The KG agent discovers tools from the full graph, while the text baseline is exposed to the full executable registry directly, subject to provider-specific limits on the maximum number of tools that can be supplied to the model. This regime stresses context saturation and retrieval ambiguity. GPT-5 is especially informative because the text baseline was capped at 128 tools by the provider, while the KG pipeline could still search over the full tool pool and compress it to a small candidate set before passing tools to the model.

## 6.3. Models and Metric

The reported runs cover Claude 4.6 Sonnet, Claude 4.6 Opus, GPT-5, and Gemini 2.5 Pro. We report mean coverage on a $[0, 1]$ scale, because it is more sensitive than pass rate to partial task completion and near misses. MCP-Atlas defines a task as passed when coverage is at least 0.75, which is appropriate for leaderboard ranking, but mean coverage is the more informative measure here because the intervention affects discovery quality before it changes binary task success. We also analyze task outcomes, tool-use

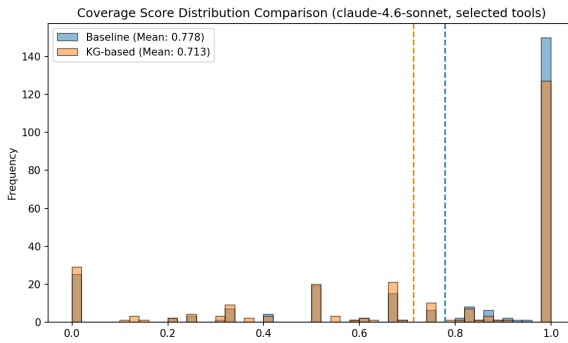 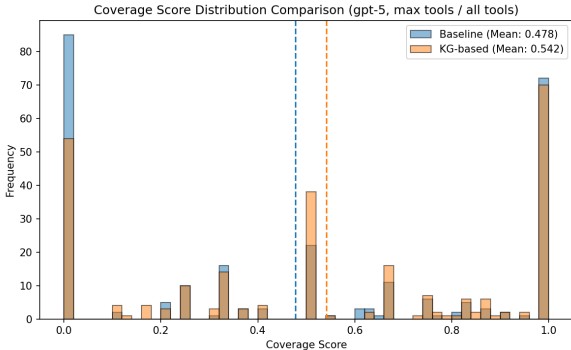

(a) Coverage histogram comparison for Claude 4.6 Sonnet.

(b) Coverage histogram comparison for GPT-5 in the all-tools setting.

**Figure 4:** Task-level coverage distributions for the two most discussion-relevant experimental settings. Panel (a) shows Claude 4.6 Sonnet, while panel (b) shows GPT-5 under all-tools exposure.

traces, and candidate-set sizes in order to explain not only whether the KG helps, but why. Where relevant, we interpret these results explicitly using the precision and recall definitions from Section 3.

## 7. Results

Table 1 reports the mean coverage scores from our MCP-Atlas runs, and Figure 4 shows the distribution of coverage for two models, Claude 4.6 Sonnet and GPT-5, in the most discussion-relevant settings. The key observation is that the KG is not uniformly superior to direct text and JSON exposure. In the selected-tools setting, the text baseline outperforms the KG for every tested model, suggesting that when the tool space has already been narrowed sufficiently, direct exposure of tool descriptions and schemas remains easier for frontier models to exploit than an additional discovery layer that can introduce recall loss.

At the same time, the all-tools setting reveals a complementary picture. For tool overload in the all-tools setting, GPT-5 improves under KG-based filtering, moving from 0.478 to 0.542 mean coverage. This improvement arises in a setting where the text baseline is constrained by a provider-specific cap of 128 tools, whereas the KG can search over the full registry of approximately 269 tools and still return only a small candidate set. As a result, KG-based filtering can recover useful tools that do not fit into the baseline's fixed tool budget. In terms of the formalism in Section 3, the KG here trades a modest decrease in $\text{Rec}_D(x)$ for many tasks against an increase in $\text{Prec}_D(x)$ and a wider effective search space for others, in a way that yields a net gain in expected coverage for GPT-5 under overload.

Raw model capability remains a dominant factor throughout. Claude 4.6 Opus is the strongest model in both settings irrespective of the retrieval strategy, while Gemini 2.5 Pro performs poorly regardless, meaning that the KG should be viewed as a complement to, rather than a standalone substitute for, strong tool reasoning. The contrast between Claude variants and Gemini is consistent with the intuition that models differ substantially in how effectively they can exploit structured metadata, and that retrieval improvements cannot fully compensate for weak tool-use capabilities.

The Claude 4.6 Sonnet all-tools condition is especially informative because task-level analysis is available for that run. In that setup, both approaches had access to approximately 270 tools in principle, but the KG reduced the discovered set to about 4.6 tools on average. Despite this roughly 98% reduction in candidate tools, the KG still achieved 0.575 mean coverage versus 0.645 for the text baseline, or about 89% of the baseline's performance. This trade-off shows that structured filtering can preserve most of the baseline's effectiveness even while dramatically compressing the search space. For systems deployed with tight tool budgets, high latency, or a need for interpretable retrieval, that may be a worthwhile exchange even before the KG surpasses the text baseline outright.

## 7.1. Why and When the KG Helps

Task-level inspection points to several recurring scenarios in which the KG helps. One scenario involves name and namespace ambiguity. When many tools overlap lexically, the text baseline can be misled by superficial name similarity or naming conventions, especially when multiple servers expose similarly named search or query tools. Capability- and server-aware retrieval helps by grouping tools semantically rather than relying only on surface form, and by steering the agent toward the correct server namespace.

A second scenario involves backend overload and wrong-source confusion. In large tool pools, the text baseline sometimes chooses the wrong backend, for example using MongoDB where Airtable or Notion would be more appropriate, or defaulting to web search when a specialized server exists. The KG helps by steering retrieval toward a capability-consistent region of the tool space that is also aligned with the expected data source, thereby increasing $\text{Prec}_D(x)$ for the discovery policy.

A third scenario arises specifically in the GPT-5 all-tools condition. Because the text baseline is constrained by a provider limit of 128 tools, it must present a truncated subset of the full registry to the model. In contrast, the KG-based pipeline can search over the entire registry of approximately 269 tools and then compress the result to a small candidate set within the same tool budget. In effect, the graph serves as a pre-filter that lifts the effective search space while still respecting the downstream budget. The observed improvement for GPT-5 should therefore be interpreted as the combined effect of structured retrieval and an expanded upstream search space. A stronger text-only baseline that also uses embedding-based retrieval over tool descriptions within the same budget would make this comparison sharper, and we consider that an important next step.

Beyond these performance metrics, the KG also offers interpretability benefits that are not directly captured by coverage scores. The discovery process can be inspected as a sequence of graph traversals and capability matches. This exposes failure modes, such as misrouted capabilities or missing taxonomy entries, and makes it easier for system builders to debug and refine tool registries in large deployments. Although we do not quantify interpretability in our experiments, practitioner experience suggests that this transparency is valuable in its own right when debugging agentic systems.

## 7.2. Why the KG Still Loses in Many Cases

The same error analysis also identifies why the KG underperforms in the other conditions. The most important factor is the recall bottleneck. Discovered tools are almost always correct, with discovery precision often around 98% or higher in the selected-tools setting, but discovery recall is substantially lower. In terms of the formalism from Section 3, $\text{Prec}_D(x)$ is consistently high, but $\text{Rec}_D(x)$ remains low for many tasks. This means the KG often returns a clean but incomplete tool set that omits at least one tool needed for full task coverage.

Capability assignment errors and taxonomy gaps steer the graph incorrectly. The most frequent issue in the Claude 4.6 Sonnet all-tools analysis was wrong data-source routing, especially when the graph steered tasks toward MongoDB or Airtable even though the task depended on local CSV files or file-based data. This indicates a failure of semantic coverage and routing quality in the current graph. When the capability taxonomy does not include a sufficiently specific class for a particular tool, or when a tool is mapped only to a very general capability, the discovery policy may stop too early or wander into the wrong subgraph, thereby excluding the correct tool from $R_D(x)$.

Some losses are implementation bugs rather than conceptual limits. Task traces expose fixable sources of failure such as a TwelveData parameter-wrapper mismatch, the wrong filesystem base path, and under-routing for Airtable. These issues likely understate the KG's eventual ceiling, because they could be addressed through better engineering and testing without changing the core ontology. Distinguishing between such engineering bugs and conceptual limitations of the KG representation is important when interpreting the current results.

The divergent tasks are also qualitatively harder. They involve cross-domain tool chaining, file system exploration, computation, and multi-step planning. This matters because it suggests that the KG's missing recall is most costly on precisely the tasks where semantic abstraction should matter

**Table 2**

Task-level outcome breakdown for the Claude 4.6 Sonnet all-tools condition.

| Outcome | Main observations |
| --- | --- |
| KG wins (56 tasks, 22%) | Better backend selection under overload, fewer naming confusions, fewer failures from web-search blocking and security-rule issues. |
| Tie (119 tasks, 46%) | Mostly simpler single-tool tasks such as weather, museum search, or direct file reads where both approaches succeed. |
| Text wins (83 tasks, 32%) | Wrong data-source steering, asking the user for file paths instead of discovering them, upstream HTTP 500 errors, and schema mismatches. |

most. In other words, the current graph captures much of the low-hanging structure for simpler tasks, but its limitations become visible on complex workflows, particularly where missing output semantics, dependencies, or side effects would be needed for robust planning.

## 8. Discussion

The results reveal an interpretation of KG-based tool discovery beyond average-score comparison. Firstly, the KG is best understood as a filtering mechanism but can be further explored for better discovery. Positive results appear exactly in the regime where the text baseline is overloaded and constrained by a hard tool budget. This is consistent across GPT-5 all-tools result and the Claude 4.6 Sonnet, that the KG compresses roughly 270 tools to 4–5 candidates while preserving most of the baseline's task coverage. In terms of Section 3, the KG improves $\text{Prec}_D(x)$ while keeping $\text{Rec}_D(x)$ sufficiently high to maintain competitive mean coverage in settings where the flat baseline cannot consider all tools.

Secondly, the main structural benefit is semantic factorization. A flat registry forces the agent to reason over tool names and descriptions directly. The KG introduces intermediate abstractions such as capabilities, parent–child relations, and server provenance, which reduce candidate entropy and make tool selection more interpretable. In effect, the KG transforms tool retrieval from full registry search into inspecting a few local candidates within a semantic neighborhood, enabling it to act as a structure-aware compression layer that preserves task-relevant tools while reducing redundancy.

Third, high precision is not enough: a graph can still lose badly if it filters out a necessary tool. The current system avoids many irrelevant candidates, but because the graph includes only capability-mapped tools and some mappings are incomplete, it often narrows the search space too aggressively. This is why the KG underperforms in curated settings, where the text baseline already enjoys manageable context and can afford broader exploration. Addressing this recall limitation requires a combination of better capability assignment, more complete taxonomies, and hybrid retrieval strategies that fall back to text or embedding-based retrieval when graph-based recall is predicted to be low.

A fourth observation is that model behavior and graph quality interact strongly. In our runs, Claude 4.6 Opus inspects the KG far more aggressively during discovery than Gemini 2.5 Pro, with many more detail-inspection calls. This suggests that the KG is not a plug-in improvement that works identically for every model: a model must still know when to inspect the graph, when to traverse further, and when to stop filtering and act. Models that are already strong at tool use and schema interpretation seem better able to exploit the additional structure, whereas weaker models may not derive much benefit and can even be harmed by the additional indirection.

The current comparison contrasts KG-based retrieval against a straightforward text baseline that exposes tool descriptions and JSON schemas as unstructured context, whereas many large-scale tool selection systems rely on embedding-based retrieval over tool descriptions. A more comprehensive comparison would therefore include at least one embedding-based baseline that uses vector search or semantic similarity to retrieve candidate tools within the same budget. Such a baseline would help separate the benefits of explicit graph structure from those of any retrieval mechanism that reduces

the search space before inference, and we expect the KG and embedding-based approaches to be complementary (e.g., embeddings within capability clusters or the KG as a scaffold for refined search).

Finally, the current ontology focuses on static tool metadata and capabilities, while information about outputs, execution constraints, dependencies, and side effects is not represented. The experiments highlight the importance of these aspects for multi-step planning and tool composition: several failures involve tasks where the model must reason about tool consequences, infer intermediate artifacts, or chain tools in specific orders. Without output and side-effect representations, the graph can support tool selection but offers limited help for compositional planning. Extending the ontology to capture output types, preconditions, and soft constraints, and aligning these semantics with planning-capable agents, is therefore a promising direction. Practically, the large reduction in candidate tools in the all-tools setting implies fewer tool descriptions and schemas sent to the model, reducing token usage and latency in many deployments. Together with the interpretability benefits of graph-based retrieval, these efficiency gains strengthen the case for KGs in tool-heavy environments, even when mean coverage is comparable to or slightly below the text baseline.

## 9. Conclusion and Future Work

We presented a knowledge graph-centered approach to representing and retrieving agentic tools, instantiated on MCP-Atlas with a focus on MCP servers and an additional overload condition derived from the same environment. The ontology models tools, servers, capabilities, and parameters, and uses the graph structure to prioritize discovery over full execution semantics. We intentionally omit output schemas, preconditions, side effects, or long-horizon composition patterns in the current model. This omission matters for the hardest cross-domain tasks, where success depends on more than choosing the right entry point and where planning over tool outcomes becomes crucial.

Empirically, our tools knowledge graph does not consistently outperform text and JSON descriptions for tool discovery, mainly due to limited graph coverage and capability assignment quality rather than the KG formalization itself. Missing assignments reduce recall even when retrieved tools are precise. However, KG-based retrieval becomes valuable under tool overload, where flat textual exposure exceeds the model's practical tool budget. In such settings, the KG acts as a structure-aware compression layer that recovers inaccessible tools and reduces agent overload.

These findings motivate several concrete directions for future work. First, improving recall through better capability assignment is a high priority. This includes designing more expressive capability taxonomies, developing robust multi-label classifiers or LLM-based taggers with explicit quality control, and integrating feedback from task failures into an iterative refinement loop for the graph. Second, hybrid retrieval strategies that combine KGs with embedding-based retrieval over tool descriptions and schemas should be explored. For example, embeddings could be used inside capability-specific subgraphs, or as a fallback when graph-based recall is predicted to be low. Third, the ontology should be extended to represent tool outputs, execution constraints, dependencies, and side effects in a way that is both tractable and expressive enough for planning-capable agents.

Finally, evaluating KGs for tool discovery on additional dimensions such as token consumption, latency, and interpretability is important for deployment-oriented scenarios. Our experiments already suggest that KGs can drastically reduce the number of candidate tools, which is likely to reduce the number of tokens needed to represent tool menus. Instrumenting these efficiency metrics and comparing against both text-only and embedding-based baselines would provide a more holistic picture of the trade-offs. Additionally, applying the ontology and retrieval pipeline to live MCP deployments or other large tool registries beyond MCP-Atlas would test their robustness in truly open-ended environments.

## Declaration on Generative AI

During the preparation of this manuscript, the authors used generative AI tools for outline brainstorming and language editing. No figures, tables, experimental designs, or quantitative analyses were

generated by AI systems. All AI-assisted content was reviewed and edited by the authors, who take full responsibility for the text and results. According to the CEUR-WS GenAI Usage Taxonomy, this represents limited use for drafting and editing assistance. No AI tools are listed as authors, and all intellectual contributions remain attributable to the human authors.

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
