# OpenReview forum: "Representing Agentic Tools in Knowledge Graphs for Structure-Aware Tool Discovery Under Tool Overload"
_ijcai.org/IJCAI-ECAI/2026/Workshop/GENAIK-NORA — IJCAI-ECAI 2026 Joint Workshop on GENAIK and NORA_

### Official Review · Reviewer_dXCx · 2026-06-05
**Knowledge Graphs for Tool Discovery: Promising Results Under Tool Overload but Recall Limitations Remain**

**Rating:** 6
**Confidence:** 4

**Review:**

Summary:
This paper studies whether knowledge graphs can improve tool discovery for LLM agents in settings with a large number of available tools. The authors build a lightweight ontology around the MCP ecosystem and represent tools through entities such as servers, capabilities, and parameters. Experiments on MCP-Atlas compare the proposed graph-based retrieval pipeline against a text-based baseline under both curated tool subsets and a large-scale tool overload scenario. The results suggest that graph-based retrieval does not consistently outperform text-based approaches, but can improve tool coverage under severe tool overload by reducing the effective search space.

Strengths:
The paper addresses a practical problem that is becoming increasingly important as agent ecosystems continue to grow. Tool overload is a real challenge for both research and deployment, and MCP provides a concrete setting in which to study it. I also appreciate that the paper presents a nuanced picture of the strengths and limitations of knowledge-graph-based retrieval rather than arguing for universal superiority. The main takeaway—that structured representations become more useful as the number of candidate tools grows—is interesting and supported by the experiments. In addition, the error analysis is informative. The discussion of capability assignment failures and routing mistakes helps explain where the current approach succeeds and where it breaks down, which makes the paper more useful for future work in this area.

Weaknesses:
My main concern is the significant recall loss introduced by the capability filtering stage. In the selected-tools setting, recall drops substantially, suggesting that the graph representation is highly dependent on accurate capability assignments. As a result, tools that are potentially relevant but not connected through the predefined taxonomy are excluded early in the retrieval process. This appears to be one of the main reasons for the reduced execution success on more challenging tasks.
I am also not fully convinced that the improvements observed in the all-tools setting are entirely attributable to the graph representation itself. The text baseline is constrained by provider-specific limits on the number of tools that can be exposed to the model, which may make the comparison less informative than intended. It would be useful to understand how the proposed approach compares against stronger retrieval methods, such as embedding-based retrieval over tool descriptions, which are commonly used in large-scale tool selection systems.
Finally, the ontology currently focuses on static tool metadata and capabilities. Information about tool outputs, execution constraints, dependencies, and side effects is not represented. While this limitation is acknowledged by the authors, these aspects are often critical for multi-step planning and tool composition, and their absence limits the conclusions that can be drawn about more complex agent workflows.


Questions:
How were capability assignments created for the MCP tools? Were they manually annotated, generated with LLM assistance, or produced through a hybrid process? Some discussion of annotation quality or assignment error rates would help clarify how much of the observed performance is influenced by taxonomy construction.
Have you compared the proposed KG-based filtering approach with embedding-based retrieval methods using the same tool corpus? Such a baseline would provide a clearer picture of whether the gains come from structured reasoning over the graph or simply from reducing the candidate set before inference.

---

### Official Review · Reviewer_TviR · 2026-06-05
**Structure-aware tool discovery**

**Rating:** 7
**Confidence:** 4

**Review:**

This paper proposes a lightweight ontology for MCP tools to aid in the representation and retrieval of tools. The work has some novelty in its focus on using KGs to represent the structure of tools for agentic solutions. It does a good job describing the approach, highlighting the contributions, and positioning the work against the current state of the art. The MCP-Atlas benchmark is used for evaluating the approach in a realistic setting, i.e., considering multiple tool calls and not directly naming the tools needed for a task.

Overall, the authors propose a good idea that seems like a natural evolution for MCP, particularly considering tasks such as disambiguation when agents are provided with multiple tools. The approach also seems to scale quite well to real-world use cases. However, the experiments only focus on measuring the mean coverage of the proposed approach against text-only tool descriptions. On this metric, the results don't favour the proposed approach across all settings but one. Personally, I think it's a missed opportunity not to include other dimensions in the evaluation, such as token consumption, since the lower number of tokens required translates into lower costs for solving the downstream tasks. Also, fewer tokens should improve latency and other metrics such as time to first token --- when compared to giving only the tools text data. Similarly, better interpretability is also mentioned in the conclusions, but it's not really discussed or evaluated in the experiments.

### Minor comments
- In the introduction, the last sentence of the first paragraph "[...] several tasks have been evolved over the last decades concerning efficient graph representation" requires a reference.
- In Section 3, the equation $E_{sub} \subset \mathcal{C}\times \mathcal{C} \rightarrow\rightarrow \text{capability-parent links}$ has two right arrows. Should this be only one?
- Prediction and Recall metrics are defined in Section 3, but they are not really used anywhere in the experiments unless I missed something.
- In Section 3, the last sentence: "This pos assists achieve good at filtering out" should be revised.
- In the Conclusion, it's mentioned that "GPT-5 improves under KG-based filtering"; however, it's not very clear what the reason for this is. Could the authors discuss more on this?
- In the Conclusion, the sentence: "meaning that the KG should be is not a standalone" needs to be revised.
- In Section 7.2, "This this indicates" needs to be revised.

Although results are not positive, I believe this work is worthy of publication in the workshop, with clear room for improvement in the experiments.

---

### Official Review · Reviewer_4FPL · 2026-06-06
**Good work on knowledge-based AI tool representation**

**Rating:** 6
**Confidence:** 4

**Review:**

Summary: This paper compares the impact of unstructured and knowledge-based tool representations on agentic AI performance. The results show that the knowledge-based approach had competitive coverage scores compared to the unstructured baseline but had poor recall despite high precision. These findings suggest that the knowledge-based approach could be useful in resource-constrained scenarios by reducing the LLM reasoning overhead.

Strengths:
1. A well thought through tool ontology is presented.
2. The paper contains a detailed analysis of the trade-off between precision and recall for the knowledge-based approach and discusses the implications for large systems.
3. The writing is easy to follow and clearly articulates the strengths and weaknesses of each approach.

Weaknesses:
1. JSON is the default format used to represent tools but was not considered in the experiment. The paper must compare JSON to the proposed ontology to be more convincing.
2. The authors are encouraged to propose methods to address weak recall in the discussion.

General feedback: Overall, this is an interesting paper, but the missing JSON experiment makes it only marginally above the acceptance threshold.

---

### Decision · Program_Chairs · 2026-06-10

Accept